# Lipid Rafts: The Maestros of Normal Brain Development

**DOI:** 10.3390/biom14030362

**Published:** 2024-03-18

**Authors:** Barbara Viljetić, Senka Blažetić, Irena Labak, Vedrana Ivić, Milorad Zjalić, Marija Heffer, Marta Balog

**Affiliations:** 1Department Medical Chemistry, Biochemistry and Clinical Chemistry, Faculty of Medicine Osijek, Josip Juraj Strossmayer University of Osijek, 31000 Osijek, Croatia; bviljetic@mefos.hr; 2Department of Biology, Josip Juraj Strossmayer University of Osijek, 31000 Osijek, Croatia; ilabak@biologija.unios.hr; 3Department of Medical Biology and Genetics, Faculty of Medicine Osijek, Josip Juraj Strossmayer University of Osijek, 31000 Osijek, Croatia; vedrana.ivic@mefos.hr (V.I.); mzjalic@mefos.hr (M.Z.); mheffer@mefos.hr (M.H.); mbalog@mefos.hr (M.B.)

**Keywords:** lipid rafts, neurodevelopment, evolution, synaptogenesis, myelination, environmental factor

## Abstract

Lipid rafts, specialised microdomains within cell membranes, play a central role in orchestrating various aspects of neurodevelopment, ranging from neural differentiation to the formation of functional neuronal networks. This review focuses on the multifaceted involvement of lipid rafts in key neurodevelopmental processes, including neural differentiation, synaptogenesis and myelination. Through the spatial organisation of signalling components, lipid rafts facilitate precise signalling events that determine neural fate during embryonic development and in adulthood. The evolutionary conservation of lipid rafts underscores their fundamental importance for the structural and functional complexity of the nervous system in all species. Furthermore, there is increasing evidence that environmental factors can modulate the composition and function of lipid rafts and influence neurodevelopmental processes. Understanding the intricate interplay between lipid rafts and neurodevelopment not only sheds light on the fundamental mechanisms governing brain development but also has implications for therapeutic strategies aimed at cultivating neuronal networks and addressing neurodevelopmental disorders.

## 1. Introduction

The development of the human brain is a complicated and dynamic process that lays the foundation for cognitive, motor, emotional and social abilities. This complex phenomenon begins in utero and extends into early adulthood [1]. It involves a variety of structural and functional changes that are influenced by both genetic and environmental factors [2]. Neurogenesis, the formation of neurons, initiates the developmental journey and sets the stage for subsequent phases, such as migration, in which neurons migrate to their destined regions in the brain [3]. This precise orchestration ensures that the basic architecture of the brain is formed. Following neurogenesis, the proliferation of synapses, or synaptogenesis, marks a period of exuberant connectivity that facilitates the initial wiring of the brain’s neuronal circuits [4]. This phase is particularly characterised by its rapid pace in the first years of life and highlights the increased plasticity and susceptibility of the developing brain to environmental factors. In parallel, synaptic pruning refines these connections by removing excess neurons and synapses to improve neuronal efficiency [5]. This process continues into adolescence and is crucial for optimising brain function. Myelination, the process by which nerve fibres are insulated with myelin, accelerates signal transmission between neurons and is crucial for the efficient functioning of neuronal networks. This process begins prenatally and continues into young adulthood [6]. This underscores the longevity of brain development and the gradual maturation of cognitive and motor functions. The concept of critical and sensitive periods highlights the windows of time when the developing brain is particularly receptive to certain environmental stimuli and emphasises the importance of timely and appropriate experiences for optimal development [7]. The cognitive, emotional and social areas each have their own trajectories and milestones that are influenced by the interplay of genetic predispositions and environmental conditions [1].

Lipid rafts, specialised microdomains in cell membranes, have been shown to be crucial regulators in the intricate process of brain development [8]. These dynamic structures, characterised by their specific lipid composition and protein content, orchestrate a variety of signalling pathways that are critical for neuronal function and development [9]. The brain, with its complex network of neurons and glial cells, relies on the precise coordination of signalling processes for its development, and lipid rafts play a central role in this sophisticated orchestration [10]. The involvement of lipid rafts in brain development involves several fundamental processes, including neurogenesis, neuronal migration, differentiation and synaptogenesis [11,12]. By compartmentalising cell membranes, lipid rafts create localised environments that facilitate efficient signal transduction, which is crucial for the spatial and temporal regulation of developmental signals. Thanks to this unique property, lipid rafts can serve as platforms for bundling signalling molecules, enhancing the specificity and speed of signal propagation essential for coordinated nervous system development [13]. In addition, lipid rafts are involved in modulating the function of neurotransmitter receptors and the organisation of synaptic components that are critical for synaptic plasticity and the assembly of functional neuronal circuits [11]. The dynamic nature of lipid rafts combined with their ability to interact with cytoskeletal elements also contributes to the morphological changes that neurons undergo during their development, such as axon formation, dendritic branching, and spine maturation. Recent advances in molecular biology and imaging techniques have shed light on the composition, dynamics and functional role of lipid rafts and provided insights into their involvement in neurodevelopmental processes [14,15]. In addition, numerous studies have been carried out in vitro on cell cultures, which represent an important tool for research into neuronal circuits [16]. Neuronal cell cultures provide precise insights into neural development by culturing neuronal networks outside the organism. This technique enables controlled experiments on neural behaviour, connectivity and the effects of various factors on neural development and provides a detailed understanding of complex neuronal dynamics [17]. However, the precise mechanisms by which lipid rafts influence brain development and the effects of their dysfunction in neurodevelopmental disorders are still the subject of active investigation.

In this article, we provide an overview of the multifaceted role of lipid rafts in brain development and highlight their contribution to important developmental processes such as neuronal differentiation, synaptogenesis, cell signalling dynamics and myelination. In addition, the influence of some environmental factors, such as dietary lipids and external factors, on the composition and functionality of lipid rafts during neuronal development is described. Understanding the interplay between genetic predispositions, environmental influences and lipid raft dynamics may shed light on their contribution to various neurodevelopmental outcomes and open new perspectives on brain development.

## 2. Structure and Function of Lipid Rafts

Lipid rafts are nanoscale, dynamic, essential microdomains of lipids and proteins that exist in the cell membrane and play a multifaceted role in cellular processes [18]. The structure of lipid rafts is dynamic and can change in response to various stimuli [19]. This dynamic organisation is crucial for the regulation of cellular processes as it enables the rapid assembly and disassembly of signalling complexes within the rafts [20]. The term “lipid raft” characterises the cholesterol- and sphingolipid-rich microdomains within cell membranes [21], which are functional units of neuronal cell membranes [22]. Cholesterol, an essential component of the plasma membrane, is crucial for the proper functioning of the nervous system and plays an important role during development and adulthood [22,23]. Cholesterol synthesis is active in the central nervous system (CNS) during the first weeks after birth, indicating its importance in different cellular processes related to neurodevelopment, such as neurite outgrowth, glial cell proliferation, synaptogenesis and myelination [22,24,25]. Sphingolipids are another important component of the lipid rafts and are essential for the development and maintenance of the functional integrity of the nervous system [26,27]. Common subclasses of sphingolipids include ceramides, sphingomyelins, glycosphingolipids (including cerebrosides and gangliosides) and sphingosine-1-phosphate. Each subclass has distinct structural features and biological functions that contribute to the overall diversity and complexity of sphingolipid metabolism and signalling in cells [28,29,30,31]. Glicerophospholipids in lipid rafts often have saturated acyl chains. They interact with cholesterol and sphingolipids and contribute to the formation and stability of these microdomains in cell membranes [18]. Their specific interactions and organisation contribute to the dynamic properties of lipid rafts and their role in various cellular processes such as signal transduction, membrane trafficking and protein sorting [32]. Lipid rafts serve as platforms for the localisation and organisation of various proteins. These proteins are crucial for the structural integrity and functional properties of lipid rafts and play diverse roles in cellular processes [33,34]. Proteins in lipid rafts that are important for neurodevelopment include receptor tyrosine kinases (Trk family receptors), G protein-coupled receptors (neurotransmitter receptors), Src family kinases (Fyn and Lyn), adaptor proteins (PSD-95 and SAP97), cholesterol-binding proteins (caveolins and flotillins), signalling proteins (Ras-MAPK, PI3K-Akt and PKC pathways) and transporter proteins. Protein kinases play a crucial role in the regulation of dendritic growth and plasticity through their precise phosphorylation of specific substrates. Dysfunctions in kinase activity are closely associated with neurodevelopmental and psychiatric disorders [35]. Changes in the expression and function of G protein-coupled receptors (GPCRs) are generally observed during ageing. These changes particularly affect the central nervous system (CNS), leading to decreased brain function, the inhibition of neuroregeneration and increased susceptibility to neurodegenerative diseases like Alzheimer’s and Parkinson’s [36]. Based on results relating to changes in the composition of lipid rafts, Alzheimer’s disease could be considered a plasma membrane disorder [37,38]. In the case of Alzheimer’s disease, modifying the lipid composition and structure of cell membranes through the depletion of sphingolipids or cholesterol holds promising potential for treating chronic inflammatory, neuropathic or cancer-related pain, particularly by targeting peripheral mechanisms [39]. Their dynamic structural organisation and functional properties make lipid rafts crucial components of neuronal membranes, influencing neuronal migration, synaptogenesis, myelination, and neurotransmission. Although the functional importance of lipid rafts in animal brain development has not been systematically studied, largely due to various challenges, understanding the structure and function of lipid rafts in neurodevelopment provides valuable insights into the mechanisms underlying normal brain development and may offer potential therapeutic targets for neurodevelopmental disorders. The most important lipid raft structures involved in neurodevelopment are shown in Figure 1.

## 3. Lipid Rafts Evolution

In this section, only the most important findings on the evolution of the molecules that are crucial for the formation of lipid rafts and their importance for brain development are addressed. Cholesterol is a hallmark of the eukaryotic plasma membrane, with a content of 20% to 50% [40]. It is also an essential component of lipid rafts, but the physical properties of cholesterol are critical for lipid raft formation and function. An appropriate range of cholesterol concentrations is essential for the rafts to exhibit certain physico-chemical properties (e.g., a certain viscosity, lipid bilayer thickness and elastic rigidity) [41]. The synthesis of one molecule of cholesterol requires eleven molecules of oxygen [42], suggesting that the evolution of the cholesterol biosynthetic pathway (CBP), which involves more than 20 enzymes, was likely an adaptive response to increased oxygen levels on Earth [43]. The emergence of the CBP coincided with the appearance of the Last Eukaryotic Common Ancestor (LECA), enabling cell function over a broad temperature scale [44]. Despite the common origin of the biosynthetic pathway, the different eukaryotic kingdoms—animals, plants and fungi—use different sterols: cholesterol (27 carbon atoms), phytosterol (28 carbon atoms) and ergosterol (29 carbon atoms), respectively [44]. The absorption of dietary phytosterols and ergosterols competes with and reduces the absorption of cholesterol in human intestines; however, neither sterol is directly converted into cholesterol [45,46]. Recent genomic studies confirm that most animals are cholesterol prototrophs capable of de novo synthesis. However, the basal eumetazoans and most species within the protostome clades (arthropods, nematodes—including *C. elegans*) are cholesterol auxotrophs, primarily due to the loss of three crucial enzymes: farnesyl-diphosphate farnesyltransferase 1 (FDFT1), squalene monooxygenase (SQLE) and lanosterol synthase (LSS) [47,48]. The independent loss of CBP genes in various animal clades can be attributed to the lack of exogenous sterols and disturbances in marine oxygen supply. In this context, the loss of energetically demanding metabolic reactions (18 moles of acetyl-CoA, 36 moles of ATP and 16 moles of NADPH per mole of cholesterol) was a selective advantage aimed at the conserving limited resources [49]. In contrast, CBP genes are conserved in deuterostomes, particularly in highly evolved vertebrates, suggesting a greater requirement for cholesterol that cannot be met from external sources [47]. We hypothesise that the independence of higher vertebrates from external sources of cholesterol was an advantage in building a complex nervous system because it is the nervous system that suffers the most consequences in terms of CBP gene mutations [50]. The role of cholesterol in early brain neurogenesis is reflected in its involvement in the Sonic Hedgehog (SHH) signalling pathway. The processing of SHH, as well as signal reception in HH-responding cells, is directly dependent on cholesterol [51,52,53,54]. We can assume that the components of the SHH signalling pathway serve as sensors of available cholesterol levels.

In vertebrates, an additional selective advantage is the uncoupling of cholesterol synthesis from extraneural tissues in the central nervous system (CNS). While most of the body’s cholesterol requirements are met by de novo synthesis in the liver (70%) or by diet (30%) [55], the blood–brain barrier (BBB) prevents the passage of lipoproteins (cholesterol carriers) into the CNS.

Before the BBB is closed, the brain utilises external sources and its own synthesis of cholesterol, both in glial cells and in neurons. The BBB function is established in the mouse brain at E10-11 [56], whereas in the human brain, this probably occurs after the 19th week of pregnancy [57]. The maturation of the BBB leads to an uncoupling of cholesterol biosynthesis, resulting in increased cholesterol accumulation in the brain. Consequently, cholesterol concentration in the human brain is 7 mg/g at birth and increases to 15–30 mg/g in adulthood, while in extraneural tissue, it averages 1.5 mg/g [58]. Cholesterol plays a crucial role in myelination and is also important for the function of lipid raft-associated proteins, synaptogenesis, neuronal plasticity, differentiation, clustering of postsynaptic receptors, response to ischemic injury and regeneration processes [50,59].

In summary, cholesterol serves as a living molecular fossil that plays an indispensable role in permeability, the regulation of lipid chain order, optimisation of protein function and, more broadly, influencing the thermomechanical properties of the plasma membrane.

The second key component of vertebrate lipid rafts is sphingomyelin (SM). It accounts for 5–10% of all phospholipids in the plasma membrane of higher animals and various invertebrates, whereas it is not found at all in fungi and plants [60]. Lower animals can synthesise a hydrophobic ceramide anchor, but instead of phosphocholine, the polar head group often consists of phosphoethanolamine, aminoethylphosphonate and their derivatives [61,62]. The chemical properties of SM (chemical inertness of choline in the polar head group and its shielding effect towards the small polar group of cholesterol, the chemical robustness of the ceramide and the relatively straight shape of the hydrophobic moiety), as well as its positioning in the outer leaflet of the plasma membrane, make it an ideal binding partner for cholesterol, as studies on artificial and biological membranes have shown [63,64]. If one adds the binding of SM to the transmembrane domains of specific proteins [65] but also the necessity of SM domains for the accumulation of phosphatidylinositol-4,5-bisphosphate in the inner leaflet of the plasma membrane during cytokinesis [66], the prerequisites for the signalling function of lipid rafts in eukaryotic cells are established. The biochemical obstacle for the formation of this favourable interaction lies in the bifurcation of the metabolism after ceramide synthesis: ceramide synthesised in the endoplasmic reticulum (ER) can be converted to SM by sphingomyelin synthase (SMS) in the medial/trans-Golgi region [67] or serve as a precursor for a wide range of glycosphingolipids whose synthesis begins with glucosylceramide synthase (GCS) and glucosylceramide in the cis-Golgi region and/or the subregion of the ER [68]. It has been shown that the formation of a heterodimer with SMS and GCS leads to the upregulation of SM [69]. In contrast, the silencing of SMS1, but not SMS2, leads to the upregulation of glucosylceramide and the synthesis of other glucosylceramides [70]. Another mechanism that favours sphingomyelin synthesis is via ceramide transport proteins (CERT), which participate in the non-vesicular transfer of ceramide into the Golgi compartment where SMS is located. Data suggest coevolution between SMS and CERT [71].

Although the brain successfully synthesises sphingomyelin, it is worth emphasising that it is also found in the diet, particularly in breast milk, and that increased intake is associated with enhanced myelination and faster cognitive development [30].

The third key component of brain lipid rafts is sialic acid-containing glycosphingolipids—gangliosides. Their synthesis is a continuation of the biosynthetic pathway originating from ceramide and glucosylceramide [72]. The major brain ganglioside structures (GM1, GD1a, GD1b and GT1b) have been well conserved in the evolution of mammals and birds [73]. Brain development is accompanied by the progressive accumulation of gangliosides, as has been demonstrated in teleost fish, chickens, mice, rats and humans [74,75,76,77,78,79]. The concentration plateau is reached during maturation and begins to decline with ageing [80,81,82]. In the adult organism, neural tissues have a one to two orders of magnitude higher concentration of glycosphingolipids compared to extraneural tissues [83]. This exceeds the phenomenon of cholesterol accumulation in neural tissues. The crucial difference is in the fact that most tissues can perform de novo synthesis of glycosphingolipids, whereas this is not the case for cholesterol.

Different developmental stages are characterised by the synthesis of different gangliosides or different ratios of glycosphingolipids, which are formed via different synthesis pathways (0, a, b and c). In most vertebrates, neurogenesis is characterised by the synthesis of simpler structures (GM3 and GD3, 9-OAcGD3). Differentiation involves the synthesis of complex structures (GD1a, GT1b and GQ1b), and synaptogenesis leads to an increase in GD1a and myelination to an accumulation of GM1 [77,79,84,85]. These changes are associated with the level and pattern of expression of enzymes involved in ganglioside synthesis, the regulation of their translation, post-translational modifications and epigenetic changes, as described in the review by Robert K. Yu and colleagues [86]. In addition to different neurogenic processes with variations in ganglioside expression, fully differentiated cell types in the brain exhibit their characteristic ganglioside patterns, as excellently demonstrated in immunohistochemical studies using monoclonal antibodies [87,88,89,90]. At this point, we would like to hypothesise that gangliosides, as an evolutionary mechanism, contribute to the enormous diversity of lipid raft compositions in the brain. The spatio-temporal complexity of their expression becomes even more apparent in imaging mass spectrometry studies [14,91,92]. In their recent work, Tania C. B. Santos, Tamir Dingjan, and Anthony Futerman introduce the concept of ‘anteome’ (from Latin “ante” for before + “omic”) to emphasise the coevolution of the network of metabolic pathways leading to the expression of glycosphingolipids in the brain [93]. The concept of the evolutionary significance of glycosphingolipids in the development of brain complexity can be criticised due to the initial relatively mild phenotype observed in *B4galnt1* (encoding enzyme beta-1,4-N-acetylgalactosaminyltransferase 1 or GM2/GD2 synthase) knock-out mice which lack the synthesis of GM1, GD1a, GD1b and GT1b [94,95,96]. It is important to emphasise that the block in ganglioside synthesis is quantitatively compensated by the synthesis of remaining structures (GM3 and GD3) and the more pronounced synthesis of alternate structures (0-series glycosphingolipids), providing evidence of the flexibility and robustness of their metabolism [97]. Recent studies clearly show that disorders related to the ganabolism or catabolism of ganglioside are associated with severe neurological symptoms [98].

In summary, we can conclude that enabling the organisation of lipid rafts is an important prerequisite and evolutionary driver for the development of the nervous system in vertebrates, as shown by perturbations in the synthesis or degradation of molecules that are the primary organisers of lipid rafts: cholesterol, sphingomyelin and glycosphingolipids.

## 4. Role of Lipid Rafts in Brain Development

### 4.1. Neuronal Differentiation and Synaptogenesis

Neural differentiation and synaptogenesis are crucial processes in the development and function of the nervous system. These complicated processes involve a variety of molecular interactions in which lipid rafts play a crucial role [99]. In our research on brain development, we propose that lipid rafts act as specialised platforms for the spatial and temporal regulation of signalling pathways that are crucial for neuronal differentiation, influence the determination of neural stem cells, and contribute to the formation of distinct neuronal subtypes during brain development. Lipid rafts serve as hubs for signalling molecules involved in neural differentiation pathways, such as the Notch [100], Wnt [101] and Sonic Hedgehog [102] pathways. The spatial organisation of signalling components within lipid rafts facilitates precise signalling processes, orchestrating the neural progenitor cell fate determination [103]. One of the key signalling processes is the activation of growth factor receptors. Growth factors, such as fibroblast growth factor (FGF), epidermal growth factor (EGF) and brain-derived neurotrophic factor (BDNF), bind to their respective receptors on the cell surface [104]. This binding triggers receptor dimerisation and the activation of downstream signalling cascades. Signalling events within lipid rafts also regulate the progression of the cell cycle of neural progenitor cells and ensure the correct timing of cell division and differentiation [103]. In vitro studies suggest that lipid rafts are also essential for glial cell line-derived neurotrophic factor (GDNF) function. The GDNF receptor, including GFRα1 and Ret, resides in lipid rafts. Through a knock-in mouse model, it has been shown that GDNF’s developmental functions in the periphery depend on GFRα1 translocation into lipid rafts [105]. Lipid rafts mediate cell adhesion and migration processes that are crucial for neural progenitor cell migration and neuronal polarisation. Adhesion molecules, including integrins and cadherins, are localised in lipid rafts and facilitate cell–cell and cell–matrix interactions that are important for neural tissue morphogenesis [106]. Additionally, there is a crucial regulatory role of lipid raft microdomains in the initial adhesion and subsequent neuronal differentiation of human mesenchymal stem cells (hMSCs). Initially, lipid rafts facilitate the internalisation of cell adhesion molecules, subsequently recruiting them to various plasma membranes [107]. In addition, lipid rafts contribute to neurite outgrowth, a fundamental process in neuronal differentiation. Proteins associated with lipid rafts, such as cytoskeletal regulators, modulate the cytoskeletal dynamics that are necessary for neurite extension and guidance [108]. The prion protein (PrP^C^), located within plasma membrane lipid raft microdomains, undergoes expression modulation based on the degree of cell differentiation. It is implicated in the complex signalling pathways governing neuronal differentiation. A recent study conducted by Martellucci et al. revealed the presence of PrP^C^ in human dental pulp-derived stem cells (hDPSCs) and its involvement in neuronal differentiation. Furthermore, the study highlights the critical role of lipid raft integrity in facilitating PrP^C^-induced signalling pathways that are essential for hDPSC neuronal differentiation induced by epidermal growth factor and basic fibroblast growth factor (EGF/bFGF) [109,110].

Throughout the neural differentiation process of the multipotent embryonic carcinoma cell line (P19 cells), the flotillin protein family member (Flot2) tends to concentrate in detergent-resistant membrane fractions (DRM Frs) rich in lipid rafts, where it associates with tyrosine kinase (Fyn). This association likely leads to the phosphorylation of Flot2 during neural differentiation. Some findings suggest that the interaction between the lipid raft scaffold protein, Flot2 and acylated proteins, like Fyn and c-Src, could be pivotal for the development and function of the central nervous system [111]. Additionally, in the nervous system, either the excessive or insufficient expression of Flot1 can contribute to the development of different neurological disorders, like Alzheimer’s disease, Parkinson’s disease and major depressive disorder [112].

Neural differentiation and synaptogenesis are closely linked processes that together control the development of the nervous system. The differentiation of neurons into specific subtypes provides the structural diversity necessary for precise synaptic connections. Synaptogenesis, in turn, refines these connections through activity-dependent mechanisms, leading to the formation of functional neural circuits that are essential for sensory processing, motor control, cognition and behaviour [113].

Synaptogenesis involves the formation of synapses, specialised junctions between neurons enabling neuronal communication. Lipid rafts play different roles in synaptogenesis and influence the formation, maturation and plasticity of synapses. They are involved in the growth of the axon and its pathfinding until the axon finds and reaches its target. However, the story does not end there. Instead, lipid rafts are involved in the formation of functional synapses and their stability. Lipid rafts are important for axonal growth and guidance, most likely due to the accumulation of signalling molecules in them [114]. Some of these are the glycoprotein M6a and ephrinB receptors, which influence the neuronal cytoskeleton [115,116], particularly the actin network [11], and others are cell adhesion molecules and receptors for the guidance of signalling proteins (e.g., netrin-1) [12,117]. When an axon reaches the postsynaptic target membrane, lipid rafts provide a suitable microenvironment for the formation of a new synapse. Cholesterol appears to be the major factor in synaptogenesis. Cholesterol has been shown to induce synaptogenesis in cultured retinal ganglion cells [4]. This induction of synaptogenesis could occur via an increase in the number of lipid rafts [118]. Moreover, cholesterol seems to be the limiting factor for the formation of numerous and effective synapses [119]. Among neurotrophins, brain-derived neurotrophic factor (BDNF) has garnered considerable attention due to its wide-ranging biological functions, notably its crucial involvement in synaptic transmission and activity-dependent synaptic plasticity [120,121]. Nevertheless, the exact role of lipid rafts in the formation of functional synapses is still unknown and is the subject of intense research. To maintain synapse function, it is important that the ratios of the lipid raft components do not change [122]. For example, the depletion of cholesterol leads to a gradual loss of synapses [121], and lipid rafts lose their functionality when their lipid–protein ratios are disturbed [123].

Lipid rafts also modulate the activity and signalling of synaptic proteins by affecting their interactions with other molecules, such as ligands, kinases and phosphatases [13,124]. A decrease in cholesterol and sphingolipid levels results in the destabilisation of surface AMPA receptors and a progressive decline in both inhibitory and excitatory synapses, along with dendritic spines [125]. In raft-depleted neurons [116], the remaining synapses and spines exhibit significant enlargement. For this study, cultured hippocampal neurons were treated with inhibitors of sphingolipid and cholesterol synthesis (such as fumonisin B1, mevastatin and mevalonate) to obtain raft-depleted neurons. This underscores the critical role of lipid rafts in maintaining normal synapse density and morphology, providing insight into why cholesterol facilitates synapse maturation in retinal ganglion cells. Moreover, this association suggests a potential connection between disrupted cholesterol metabolism and the loss of synapses observed in neurodegenerative diseases [120].

In summary, lipid rafts are dynamic structures important for synaptogenesis because they provide a suitable microenvironment not only for synaptogenesis but also for the maintenance of stable and functional synapses. The disruption of lipid rafts during synaptogenesis can impair normal behaviour and result in psychiatric disorders (depression, schizophrenia, etc.) [126].

### 4.2. Myelination

Lipid rafts contribute to the regulation of myelination processes in the developing brain. By influencing the distribution of lipids, particularly sphingolipids and cholesterol, within the myelin sheath, lipid rafts can affect the structural integrity and functionality of axons. This, in turn, affects the speed and efficiency of signal transmission in neuronal circuits.

Myelination is a crucial biological process in which a myelin sheath forms around nerve fibres or axons. The main function of myelin is to insulate the axons and increase the speed at which electrical impulses (action potentials) are transmitted along the axon. This is achieved by reducing the capacitance of the axon membrane and increasing its electrical resistance. A notable benefit of myelin is its ability to reduce the energy required to transmit nerve impulses, making this process more efficient [127]. This efficient signal transmission is essential for proper neurological function, including motor control and sensory processing. In addition, myelin provides trophic support to the axon and maintains its long-term integrity. It also facilitates three-directional communication between the neuron, the myelinating cell and the environment [128].

Myelination is a dynamic process that begins during foetal development and continues into early adulthood. It is initiated when either oligodendrocytes in the central nervous system (CNS) or Schwann cells in the peripheral nervous system (PNS) recognise and bind to an axon. According to Simons et al., myelination begins with the establishment of cell polarity, a phenomenon that is triggered by external signals emanating from the axon. The assembly of the myelin sheath begins with the pre-assembly of myelin-specific proteins and lipids that undergo the biosynthetic pathway. This preparatory phase ensures that the myelin components are correctly organised before they reach the plasma membrane. Once at the plasma membrane, these pre-assembled domains are further organised and expanded through interactions with myelin basic protein (MBP). MBP plays a central role in this process. It is synthesised locally at the site of myelination by the local translation of its mRNA. The interaction of MBP with the pre-assembled myelin components promotes their assembly and thus improves the structural integrity of the myelin sheath [129].

During the formation of myelin, the axon is coated with layers of cell membranes. Oligodendrocytes expand their cell membranes to wrap around several axons, while Schwann cells myelinate a single axon segment. This sheathing process proceeds in a spiral way, and as it progresses, the layers of the cell membrane thicken to exclude most of the cytoplasm and form a dense, lipid-rich sheath [130]. As a result of this sheathing process, a compact, multilayered sheath is formed that includes characteristic regions such as the nodes of Ranvier and the internodal segments. The nodes of Ranvier are periodic gaps in the myelin sheath that occur at regular intervals along the axon. They play a crucial role in facilitating the rapid conduction of electrical impulses through a process known as saltatory conduction. In this mechanism, nerve impulses jump from one node of Ranvier (an unmyelinated gap) to the next, bypassing the myelinated segments. This sophisticated process significantly increases the speed of signal transmission in the nervous system [131].

Myelin consists mainly of lipids, which make up about 70% of its composition, while proteins account for the remaining 30%. This unique composition contrasts with most biological membranes, which have a roughly equal ratio of proteins to lipids [131]. The high lipid content in oligodendrocytes and myelin membranes includes cholesterol, phospholipids and glycosphingolipids (GSLs). Cholesterol and GSLs are the major lipid components of myelin, accounting for ~27% and 31% of total myelin lipids, respectively [130,131]. GSLs are a class of sphingolipids that have a sphingoid base, a straight-chain amino alcohol with 18–20 carbon atoms, usually carrying a saturated or unsaturated fatty acid chain. Depending on the attachment of a mono- or oligosaccharide head group to the sphingoid base, different subclasses of GSLs are formed, such as cerebrosides, sulfatides and gangliosides. Two GLSs are especially enriched in the myelin membrane, galactosylceramide, (GalC) and its sulfated derivative, sulfatide, which account for approximately 23 and 4% of the total lipid pool [132]. Although galactolipids are important key factors for the integrity and long-term maintenance of myelin membranes, their presence does not appear to be essential for myelin biogenesis and assembly [133,134]. These data clearly indicate that both GalC and sulfatide play distinct roles in oligodendroglial differentiation, myelin maintenance and the overall stability and functionality of the myelin membrane.

These lipids, together with cholesterol, are involved in the formation of lipid rafts [8,135] that, in the context of myelination, play an important role in regulating myelination processes in the developing brain [10]. These signalling domains in the cell membranes of oligodendrocytes organise receptors and signalling molecules that are required for myelination. Enhancing the production or accessibility of cholesterol and gangliosides could potentially be associated with Huntington’s disease [136].

Gangliosides consist of a sialylated glycan attached to a ceramide lipid and are mainly located on the outer surface of the plasma membrane. With their outward-facing glycans, the gangliosides associate laterally with each other, sphingomyelin, cholesterol and selected proteins in lipid rafts [73]. However, gangliosides are not always components of lipid rafts. It has been shown that the distribution between the rafts and the non-raft parts of the membrane depends on their concentration. At physiological concentrations, gangliosides can form their own domains, which are preferentially located in the liquid-disordered (non-raft) part of the membrane [137,138]. Two of the four major brain gangliosides, GD1a and GT1b, are expressed on the axonal membrane and interact with the sialic acid-binding protein MAG on the periaxonal surface of the myelin membrane [139,140]. In this way, these molecules on opposing surfaces facilitate the interaction between axons and glia and promote myelin stability [141,142]. These interactions were investigated with genetically modified mice. Depending on the mutation, the phenotypes can be very different. Mice with a disrupted GM2/GD2 synthase gene that lacked complex gangliosides showed only subtle defects in their nervous system [96], but later studies showed marked pathology in these mice, including axonal degeneration, an increase in unmyelinated fibres and redundant myelin loops, the disruption of paranodal junctions and the mislocalisation and dysfunction of ion channels [143,144]. In contrast, mutants lacking the gangliosides of the -b and -c series (GD3 synthase-null) show no demyelination in the brain [145]. The most severe phenotype was observed in the transgenic mice completely lacking ganglio-gangliosides (Siat9 and Galgt1 double knock-out). These mice showed profound axonal degeneration, the vacuolisation of CNS white matter and impaired paranodal stability, as evidenced by the presence of paranodal loops facing away from the axon [146]. These studies suggest a role for gangliosides in the interaction between axons and glia. As mentioned above, the interactions between lipids and proteins are crucial for myelin formation and the maintenance of myelin. They regulate protein transport and molecular organisation within the myelin sheath [6].

Myelin expresses a unique set of lipid raft-associated proteins, including proteolipid protein (PLP), myelin basic protein (MBP), myelin oligodendrocyte glycoprotein (MOG), myelin-associated glycoprotein (MAG), 2′,3′-cyclic-nucleotide 3′-phosphodiesterase (CNP), myelin and lymphocyte protein (MAL) and neurofascin-155 (NF155), of which PLP and MBP are the most abundant [147].

PLP is an integral membrane protein that has two different isoforms, PLP and DM20 [148]. The protein primarily plays a role in stabilising the intraperiod line by assembling the extracellular leaflets of the myelin membrane [149]. PLP contains a cholesterol recognition/interaction sequence and is a major cholesterol-interacting protein in oligodendrocytes, and these properties may be responsible for the association of PLP with these domains [150]. However, PLP mice do not show a demyelination phenotype, suggesting that PLP is not essential for the actual assembly of the myelin sheath. Instead, PLP is important for the proper assembly of myelin lipid rafts [132]. PLP may also be important at the ultrastructural level once myelin has formed, as in PLP-null mice, the extracellular compaction of neighbouring membranes is abnormally condensed, which could be a sign of reduced myelin stability [151]. Overall, it is not the absence of PLP but the expression of properly folded PLP in physiological amounts that seems to play a crucial role in the proper assembly of myelin [132].

MBP is the second most abundant protein in myelin and accounts for 30% of the total myelin protein in the CNS. MBP exists in various isoforms of different sizes and charges, which are formed by alternative splicing of an mRNA transcript [152]. MBP is the only known structural myelin protein that is absolutely required for myelin membrane formation, presumably due to its role in myelin membrane compaction, as it can hold the cytoplasmic leaflets together [153]. Due to its highly positive charge, MBP can interact as a cytoplasmic peripheral membrane protein with anionic phospholipids in the inner leaflet of the myelin membrane, and this binding is probably crucial for myelin assembly [152]. Interestingly, the dynamics of MBP are also influenced by changes in the dynamics of the galactolipids of the extracellular leaflet [154], suggesting that “indirect” interactions may occur between the galactolipids of the extracellular leaflet and MBP at the cytoplasmic surface of the membrane. Given its actin-binding properties, MBP may also play a key role in the transmission of galactolipid-mediated signalling. The destruction of the myelin sheath leads to neurodegeneration and conduction failure, as observed in demyelinating diseases such as multiple sclerosis (MS). MBP has long been investigated as a factor in the pathogenesis of the autoimmune neurodegenerative disease multiple sclerosis [155], and it is clear that MBP and its functions in myelin formation and long-term maintenance are associated with MS.

MAG is a transmembrane protein that is selectively expressed by myelinating cells [156] and is an important component of the myelin sheath. In the CNS, MAG is only found on the innermost (periaxonal) myelin sheath, which lies directly on the axon surface. This protein plays a crucial role in the interaction between myelinating glial cells and axons. It contributes to the stability and integrity of the myelin sheath by mediating adhesion between the myelin membrane and the axon. Mice lacking MAG show abnormalities in the formation of the periaxonal cytoplasmic collar, suggesting that the interaction between MAG and ganglioside is important for the organisation and maintenance of the periaxonal space and the cytoplasmic collar [157]. The phenotypes of mutant mice with altered ganglioside expression, as described above, are consistent with a role for GD1a and GT1b in MAG functions [142]. Mice lacking complex gangliosides (B4galnt1-null) show many of the same phenotypic features as Mag-null mice, including myelin abnormalities, progressive axon degeneration in the central and peripheral nervous system, reduced neurofilament spacing, reduced the diameters of myelinated axons and disrupted the nodes of Ranvier [144].

The myelin and lymphocyte protein (MAL) is associated with the lipid rafts of myelin membranes. MAL is upregulated in mature oligodendrocytes and is involved in the apical sorting machinery of polarised cells [158]. It is speculated that MAL may be involved in the intracellular targeting of neurofascin-155 (NF155) [159], a key protein required for paranodal formation [160].

In summary, myelin is an indispensable component of the nervous system, ensuring the rapid and efficient transmission of nerve impulses. The orchestration of the myelination process by specialised glial cells is fundamental to the health and functionality of neural networks. Myelin’s significance extends beyond facilitating communication between neurons; it also underpins cognitive development and contributes to the resilience and adaptability of the nervous system throughout an individual’s life. Lipid rafts play a pivotal role in the myelination process within the developing brain, serving as crucial platforms for organising signalling molecules, facilitating interactions between cells, and regulating intracellular signalling pathways. These specialised membrane domains also influence the physical properties of the membrane and support axonal contact and guidance, which are essential for the proper formation of myelin. When lipid rafts malfunction, it can result in demyelination, leading to a range of neurological disorders and impairments.

### 4.3. Environmental Influences

Our hypothesis suggests that environmental factors, including dietary lipids and external stimuli, may modulate the composition and function of lipid rafts during brain development. Understanding the interplay between genetic factors, environmental influences and lipid raft dynamics may shed light on how these microdomains contribute to individual variability in neurodevelopmental outcomes. Environmental factors such as maternal nutrition [161,162], exposure to pathogens [163] and hormonal influences [164], especially glucocorticoids through chronic stress [165] during pregnancy, can influence the composition, organisation and functionality of lipid rafts in the developing foetal brain [166].

#### 4.3.1. Maternal Nutrition

Maternal nutrition is an essential foundation for proper foetal brain development [7]. Nutrients such as cholesterol, omega-3 fatty acids and glycolipids are essential for proper neuronal membrane formation and synaptic connectivity [167,168].

Cholesterol modulates membrane fluidity, provides structural support and influences protein localisation and signal transduction through lipid rafts [9,18]. Disruption or removal of cholesterol from the raft leads to the dissociation of proteins and disrupts their function and signalling through lipid rafts [18]. Cholesterol is either supplied from the diet or synthesised de novo by the liver, intestine and skin [169], starting from acetyl-CoA as a substrate and involving at least 20 enzymes [170]. A genetic deficiency in cholesterol metabolism causes Smith–Lemli–Opitz syndrome, in which children show a range of symptoms, including learning disabilities [171]. Staneva and coworkers analysed possible raft-related consequences of the accumulation of the metabolic precursor of cholesterol, 7-dehydrocholesterol, which occurs in Smith–Lemli–Opitz syndrome, and how subtle differences in the structure of the sterol nucleus can cause membrane lipid dysfunction [172,173]. The domains containing 7-DHC were smaller and had more diffuse boundaries, and the location of 7-DHC was more loosely defined than that of cholesterol. The difference in sterol structure can play a role in membrane organisation and the sorting of certain proteins, such as the morphogenic protein Sonic Hedgehog [102]. In addition, 7-DHC is more susceptible to oxidation than cholesterol, and its accumulation can increase oxidative stress in cells, potentially exposing membranes to the dangers of free radicals. Dietary cholesterol is derived exclusively from animal products and is associated with the decreased synthesis of endogenous cholesterol, a compensatory mechanism that maintains constant cholesterol homeostasis [174].

Omega-3 (n-3) polyunsaturated fatty acids (PUFAs) such as docosahexaenoic acid (DHA) and eicosapentaenoic acid (EPA) are found in fish and some vegetable oils such as flaxseed and are integral components of cell membranes [175]. They can infiltrate into lipid rafts and increase the unsaturated fatty acids content, alter the physical properties of lipid rafts by affecting their fluidity and stability, which is incompatible with rigid cholesterol, and disrupt the organisation of lipid rafts, which could affect downstream signalling. [176,177]. The increased fluidity affects the mobility and arrangement of proteins within the rafts [178]. DHA controls neurotransmission through dopamine and serotonin [179], protects neurons from apoptosis [180], promotes growth cones during brain development [181] and regulates nerve growth factors [182]. It remains to be clarified whether the abovementioned effects are due to the influence of n-3 PUFAs on lipid rafts.

Since gangliosides are important raft residents, dietary intake of these glycolipids during pregnancy could be crucial for proper neurodevelopment. Diary, animal and soy products are rich in gangliosides and should be included in the diet during pregnancy [183]. Although dietary gangliosides are degraded in the intestinal tract, their degradation products (saccharides and lipids) are absorbed and affect ganglioside expression in tissues [184]. Compared to the peripheral organs, the brain contains more gangliosides, and their distribution is more concentrated in grey matter than in white matter [185]. Between weeks 16 and 22 of gestation, ganglioside concentration in the frontal cortex doubles, increases by 30% in the hippocampus and remains highly concentrated until 30 weeks of gestation. Among various ganglioside species, GM1 has been associated with brain function [186]. Changes in ganglioside composition during development may be a feature of brain differentiation, probably through modifications of lipid rafts [187]. The role of gangliosides in the CNS can be investigated by studying knock-out mice models lacking a particular ganglioside or with impaired glycosphingolipid metabolism. In all cases of ganglioside knock-out models, disorders involving lipid rafts have been found to be associated with CNS disorders [73,188].

The availability of these nutrients during pregnancy alters the lipid composition of lipid rafts, potentially altering their structure and function in the developing brain. In addition, dietary components, such as antioxidants, vitamins and polyphenols, can also modulate membrane properties and indirectly influence lipid raft behaviour and associated cellular processes [189].

#### 4.3.2. Pathogens

Many pathogens can be dangerous for foetal development during pregnancy, both viral (SARS-CoV-2, cytomegalovirus, varicella zoster, Zika virus [190,191] and bacterial (group B Streptococci, *Listeria monocytogenes* [192,193]. For pathogens, a lipid raft can be viewed as an “island with an airport in the middle of the ocean” equipped with various proteins/receptors that ensure that a pathogen reaches a desired location, supports pathogen replication and deceives the host immune system [194,195]. 

The protease-independent entry of SARS-CoV-2 into the cell depends on the localisation of angiotensin-converting enzyme 2 (ACE2) receptors in lipid rafts [196]. Moreover, the SARS-CoV-2 spike protein has a lateral N-terminal domain that binds to gangliosides residing in lipid rafts [197].

Cytomegalovirus (CMV) is a leading cause of non-genetic hearing loss in children and causes neurodevelopmental delay. CMV alters lipid rafts by increasing cholesterol efflux and reducing the number of ABCA1 cholesterol transporters [198].

Varicella zoster causes serious health issues, with problems ranging from skin defects to nervous system impairments. Like CMV, it utilises lipid rafts at many stages of infection. Rafts contribute to viral envelope integrity and endocytosis [199].

Zika virus, which causes microcephaly in newborns, utilises lipid rafts as an entry point into cells and alters them by accumulating flotillin-1 [200,201].

Bacterial infections can also cause many neurodevelopmental problems in babies and very often utilise lipid rafts in a similar was as viruses. Bacterial pore-forming toxins (e.g., aerolysin, streptolysin O) can interact with lipid rafts, leading to their disruption or alteration [202]. Many bacterial species manipulate raft proteins to facilitate their survival and replication or to become invisible to the immune system [99].

Newborns infected with group B Streptococci (GBS) can develop pneumonia, septicemia and meningitis. Raft-associated phosphatidylinositol 3-kinases have been shown to be involved in GBS invasion [203].

*Listeria monocytogenes* causes severe infection with high mortality and neurodevelopmental abnormalities [204]. In this case, lipid rafts provide a binding site for bacterial pore-forming toxin listeriolysin O [205].

By targeting and altering the properties of lipid rafts, pathogens can alter cellular processes, thus contributing to successful infection and propagation in the host. This can be particularly dangerous for the developing foetus, as pathogens can cross the placenta and directly influence foetal development.

#### 4.3.3. Drugs

Drugs can influence the function of lipid rafts in various ways. Some drugs interact directly with lipid components or proteins residing in lipid rafts, altering their stability or signalling, while others interfere with raft components by altering their composition [206].

Several classes of drugs have been studied for their influence on lipid rafts:Statins: These drugs are used to lower cholesterol levels by inhibiting β-hydroxy β-methylglutaryl-CoA (HMG-CoA) reductase, an enzyme involved in cholesterol synthesis. Some studies suggest that statins interfere with lipid rafts due to the hydrophobicity of the drugs. In addition, statins modulate the expression of the ACE2 receptor, which is located in the lipid rafts. It has been shown that cerivastatin increases the homogeneity of lipid rafts, making the raft aggregates smaller [207,208].Anaesthetics: Certain anaesthetics, such as propofol and isoflurane, have been reported to interact with lipid rafts by activating phospholipase D2 [209,210].Antiviral drugs: Some antiviral drugs, such as entry inhibitors [211], target lipid rafts as part of their mechanism of action. Some of those drugs aim to prevent viral entry into host cells by interfering with ACE2 receptors located in lipid rafts [212,213]. 25-hydroxycholesterol has been shown to interfere with cholesterol in the membrane, making the bilayer less rigid and affecting the composition of lipid rafts [214].Steroids: Steroid hormones, such as glucocorticoids and sex hormones, are known to interact with cell membranes, including lipid rafts [215]. Glucocorticoids remove the acyl-bound adaptor proteins and phosphoprotein required for T cell activation from lipid rafts by inhibiting their palmitoylation [216]. In a small portion of estrogen receptors (ERs), ER-α is located in the lipid rafts, exerting the effects of estrogens on lipid rafts directly through their receptors [164].Antipsychotics and antidepressants: Many antidepressants have been shown to colocalise with 5-HT_3_ receptors in lipid rafts [217]. Trazodone, an antidepressant, and aripiprazole, an antipsychotic, have been shown to alter cholesterol metabolism and, therefore, likely affect lipid raft composition and stability [218,219].

Understanding how drugs interact with lipid rafts is crucial as it can provide insights into therapeutic strategies targeting these specific membrane regions for various medical conditions and avoiding unwanted side effects.

#### 4.3.4. Hormones

Hormonal influences, including maternal stress and hormonal fluctuations during pregnancy, influence foetal brain development [220]. Some of the hormonal effects are directed towards lipid rafts.

Chronic stress during pregnancy and elevated glucocorticoid levels increase the risk of neurodevelopmental disorders such as autism, ADHD, schizophrenia and depression [221]. Although glucocorticoids can be used to promote the rapid maturation of underdeveloped organs in a developing foetus, they can cause unwanted side effects [222]. It has been observed that glucocorticoids exert their effects via their receptors in lipid rafts [223]. In addition, all the conditions mentioned have already been associated with changes in lipid rafts [224,225,226,227].

Hormonal fluctuations during pregnancy are important for proper foetal development. In the context of foetal brain development, hormones such as oestrogen, progesterone and thyroid hormones orchestrate neuronal proliferation, migration, synaptogenesis, myelination and neurotransmitter release and contribute to overall brain growth [220,228,229,230]. All of the abovementioned hormones interact with lipid rafts [164,231,232]; therefore, improper hormonal balance can influence lipid metabolism, which alters the lipid composition of cell membranes and subsequently affects the structure and function of lipid rafts in the foetal brain.

Overall, the environmental influences on lipid rafts during foetal brain development underscore the importance of maintaining a healthy maternal environment. Optimising maternal nutrition, minimising exposure to toxins, managing stress and ensuring hormonal balance during pregnancy are crucial factors that can positively influence lipid raft composition and function in the developing foetal brain.

## 5. In Vitro Neural Cell Culture—Model to Study Role of Lipid Rafts in Neurodevelopment

Understanding the intricate processes of neural development, from the formation of individual neurons to the assembly of complex neural circuits, is crucial for deciphering complex brain functions. The in vitro neural cell culture, i.e., the cultivation and maintenance of neurons outside the body, is a powerful tool for gaining insights into neural development and its underlying mechanisms [233]. The cells used for culturing neuronal or neuron-like cultures can be primary, immortalised continuous lines or derived/differentiated from pluripotent stem cells [234]. Two main methods are generally used for culturing neuronal cells: two-dimensional (2D) and three-dimensional (3D) cultures. In 2D culture, neurons are grown on flat surfaces. This approach allows for the easy manipulation and visualisation of the neurons but may not accurately reflect the complex architecture of the developing brain. Specific treatments can be applied, and the behaviour of individual cells in response to the treatment can be observed in real time. Jose et al. determined the importance of cholesterol during the development and polarisation of rat hippocampal neurons [235]. Methylbetacyclodextrin (MβCD) was used to transiently deplete cholesterol from neurons in the early developmental stage. The neurons were derived from P0/P1 rat pups. The reduction in the length and number of neurites corresponded to the applied concentration of MβCD. On the other hand, transient replenishment with cholesterol partially reversed the effects of MβCD treatment [235]. This effect of cholesterol on neuronal polarisation could only be observed in neurons that are not in direct contact with the surrounding neurons and, thus, are not affected by synaptic signalling. Another good example of the use of 2D cell cultures in neurodevelopment research is the study of neuronal growth cone morphology and development. Cells can be differentially stimulated with different multiple growth factors or compounds, and the growth cone is then monitored for changes in terms of morphology and protein expression [236]. Similar to cholesterol, very long-chain fatty acids (VLCFAs) also play an important role in neuronal polarisation in the early developmental stages. A lack of C16:0 ceramide and C24:0 phosphatidylcholine inhibits brain development in vivo. In experiments with E14.5 hippocampal cell culture, VLCFAs were found to be important for the proper integration of sphingolipid into the lipid rafts of the neuronal growth cone and its subsequent development [237]. This experiment gave a good indication of why a lack of C16:0 ceramide and C24:0 phosphatidylcholine specifically affects the early stages of brain development. In both experiments, the cells were grown at low density per area, which, in turn, prevented the neurons from forming proper networks that normally occur in the brain. Seeding cells at a high density partially alleviates this problem by forming a relatively flat synaptic network. The formation of synapses can be observed as a function of lipid raft composition and without significant obstacles. By overexpressing the lipid raft-associated protein flotillin 1 in hippocampal cell cultures, Swanwick et al. found that it plays a role in the formation of glutaminergic synapses [238]. Further experiments on primary hippocampal cell cultures showed the importance of tightly controlling lipid rafts in the formation and stabilisation of AMPA receptors. Both receptors and their association with lipid rafts are necessary for the formation of stable synapses. It is important to have a tool that is capable of rapidly monitoring the formation, disruption or enhancement of synapse formation, and densely seeded primary neuronal cell cultures provide this tool [122,125,239,240].

Growing cells on flat surfaces can reveal relatively limited information about actual neuronal organisation and interaction in the brain. Three-dimensional culture aims to recreate the three-dimensional environment of the brain by using scaffolds or hydrogels that provide a more natural substrate for neuron growth and interactions mimicking the extracellular matrix (ECM). This approach can generate better-organised neuronal networks that resemble the structure and function of the brain in vivo [16,241]. Furthermore, 3D scaffolds can be specifically modified by adding different proteoglycans or laminins to promote targeted differentiation or neurite growth [242,243,244]. The simplest form of 3D cell culture for neurons are spheroids, as no 3D scaffold is required for their formation. Rather, a hydrophobic vessel or suspension in the growth medium is required to facilitate cell clumping. In this way, a gradient of nutrients and oxygen is created, and the cells are only connected to the extracellular matrix cells they produced and neurite tangles [242,245,246]. A further development of spheroids is the suspension of cells in a scaffold such as Matrigel, which is immersed in a growth medium. In this way, the cells are held in place, and neurite growth can be selectively stimulated and observed. Most experiments using this approach aim to repair damaged brain tissue from animal models or simulate the damage in vitro [242,247,248,249]. The most complex of all 3D neuron cell cultures are brain organoids. Brain organoids have emerged as a powerful tool for studying neurodevelopment. These miniature brain models, derived from stem cells, self-organise into structures with or without ECM, reflect different brain regions, and recapitulate many aspects of early brain development. Brain organoids can be generated from a variety of stem cell sources, including embryonic stem cells (ESCs), induced pluripotent stem cells (iPSCs) and neural progenitor cells (NPCs) [17,250]. ESCs are pluripotent cells that can differentiate into all cell types of the body, whereas iPSCs are derived from adult cells that have been reprogrammed to a pluripotent state and can be harvested noninvasively from human donors. NPCs are multipotent cells that can give rise to neurons, astrocytes, and oligodendrocytes [251]. Brain organoids have been instrumental in elucidating the molecular signals that regulate NPC proliferation, differentiation, and migration, providing insights into how these processes are perturbed in various neurodevelopmental disorders. Additionally, they are a tool to study the formation of neural circuits, the intricate networks of neurons that underlie brain function. By manipulating the environment of brain organoids, the effect of various factors, such as neuronal interactions, neurotransmitter signalling, and environmental stimuli, can influence the development of neural circuits [252,253,254,255,256,257]. To date, the role of lipid rafts in brain organoid development has not been studied in depth. This provides a good opportunity to study the effects of different cholesterol levels and VLCFA derivatives on the development of appropriate neuronal networks in vitro using human cell-derived brain organoids. 

## 6. Conclusions

In summary, lipid rafts play a critical role in the development and function of the nervous system in vertebrates, as highlighted by the impact of disorders affecting key lipid raft organisers like cholesterol, sphingomyelin and glycosphingolipids. These dynamic structures not only facilitate synaptogenesis but also contribute to the maintenance of stable and functional synapses, with disruptions potentially leading to psychiatric disorders. Moreover, myelin, which is orchestrated by specialised glial cells, is essential for efficient nerve impulse transmission and cognitive development, with lipid rafts serving as crucial platforms for myelination processes. Dysfunctions in lipid rafts can result in demyelination and neurological disorders. Recognising the environmental influences on lipid rafts underscores the importance of maintaining a healthy maternal environment during foetal brain development, emphasising factors such as nutrition, toxin exposure, stress management, and hormonal balance to optimise lipid raft function and support healthy neural development. 

## Figures and Tables

**Figure 1 biomolecules-14-00362-f001:**
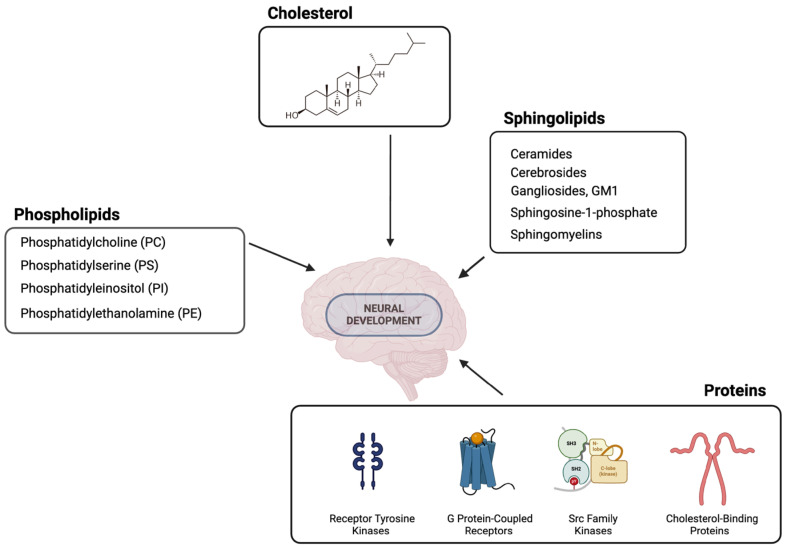
The most relevant lipid raft components important for neurodevelopment. The figure was made using https://www.biorender.com/, accessed on 18 January 2024.

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
