# Peer review of "Lipid Rafts: The Maestros of Normal Brain Development"

_biomolecules, 2024, doi:10.3390/biom14030362_

Round 1
Reviewer 1 Report
Comments and Suggestions for Authors
Report on the review entitled “Lipid rafts: the Maestros of normal brain development” by Viljetic and colleagues
The manuscript of Viljetic and co-workers provides an overview of the role of lipid rafts in brain development. They discusse in details the role of lipid raft constituents in neuronal differentiation, cell signaling mechanism, synaptogenesis and myelination. They describe the role of dietary lipids and some external factors on the functions of lipid rafts during neuronal development. They describe, that lipid rafts play an important role in the development and function of the nervous system in vertebrates and that lipid raft disruption leads to demyelination and neurological disorders. This is a well-documented review that shows the role of the components of lipid rafts in the central nervous system from several aspects. The manuscript will certainly be interesting to other scientists in the field. I have outlined my suggestions as follows:
1. The authors could highlight that lipid rafts are mentioned as potential drug targets in several publications, as summarized in a very recent review (PMID: 38290404). I suggest two other recent review papers that detailed the role of rafts in neurological disorders (PMID: 37858892, PMID: 37772385).
2. In the boxes of the figure, the structures are not visible at all, their quality is very poor, their resolution should be improved.
This manuscript is suitable to publish in Biomolecules after a minor revision.
Author Response
Dear Reviewer 1,
Thank you for taking the time to review our manuscript. Your insightful comments and feedback are greatly appreciated, and we are grateful for the opportunity to address them.
We believe that your comments provide valuable insights that will help strengthen the quality and clarity of our work.
We have carefully reviewed each of your comments and suggestions, and we are pleased to provide our responses and proposed revisions below. Your feedback is instrumental in refining our research and ensuring its significance in the field.
Answers are attached.
Sincerely,
Authors

Reviewer 2 Report
Comments and Suggestions for Authors
The manuscript by B. Viljetić and colleagues reviews the roles of lipid-protein domains (rafts) in brain development and functioning. The review seems to me very detailed and comprehensive. Various mechanisms of possible involvement of rafts in these processes is discussed; problems unsolved to date in the field are highlighted. The review is based on a huge number of relatively recent articles.
I have only several minor comments.
1) Several times over the text the authors write, that rafts have an unique composition and structure (e.g., lines 55, 133). At the same time, the authors discuss that the composition and structure of rafts can be different or even regulated (e.g., lines 95-96, 256, 538). These two statements look contradictory; this issue should be clarified or discussed. Along with direct regulation of raft composition, it may be that several types of sphingomyelin-rich domains can exist in cells. The evidence for this possibility was obtained in the work [Ayuyan, A. G., & Cohen, F. S. (2008). Raft composition at physiological temperature and pH in the absence of detergents. Biophysical Journal, 94(7), 2654-2666], where four types of domains were observed.
2) Lines 111-112: “Phospholipids in lipid rafts are mainly those that interact with cholesterol and sphingolipids ...”. Strictly speaking, sphingomyelin is phospholipid; thus, the contraposition “shpingolipids - phospholipids” is not exactly correct.
3) Lines 113-116: “Some common phospholipids in lipid 113 rafts are phosphatidylcholine (PC), phosphatidylethanolamine (PE), phosphatidylserine (PS) and phosphatidylinositol (PI). Together with cholesterol and sphingolipids, these phospholipids form the core structure of lipid rafts.” The distribution of lipids toward rafts is mainly determined by saturation of their hydrophobic chains: rafts prefer to recruit saturated lipids, and the type of the polar head is a minor factor. With respect to polar heads (PC, PE, PS, PI), rafts are definitely depleted by PE and PI lipids that have “physiological” hydrophobic chains. In model systems, rafts are shown to be strongly depleted in DOPC, the mole fraction of which does not exceed 5% [Veatch, S. L., Polozov, I. V., Gawrisch, K., & Keller, S. L. (2004). Liquid domains in vesicles investigated by NMR and fluorescence microscopy. Biophysical Journal, 86(5), 2910-2922]. Thus, the statement that “these phospholipids form the core structure of lipid rafts” seems not exactly correct.
4) Lines 193-195: “Cholesterol plays a crucial role in myelination, as it accounts for 40% of the lipid content in the membranes of myelinating cells.” This argument to underscore the importance of cholesterol for myelination seems not conclusive, as an average content of cholesterol in membranes of the most cells is about the same, i.e., 40%.
5) Lines 151-152: “Cholesterol is a crucial component of lipid rafts, representing a hallmark of the eukaryotic plasma membrane with a content of 20% to 50% [38].” According to works [Ayuyan, A. G., & Cohen, F. S. (2008). Raft composition at physiological temperature and pH in the absence of detergents. Biophysical Journal, 94(7), 2654-2666; Frisz, J. F., Klitzing, H. A., Lou, K., Hutcheon, I. D., Weber, P. K., Zimmerberg, J., & Kraft, M. L. (2013). Sphingolipid domains in the plasma membranes of fibroblasts are not enriched with cholesterol. Journal of Biological Chemistry, 288(23), 16855-16861] SM-rich rafts are not enriched in cholesterol as compared to the surrounding plasma membrane. In model membranes, cholesterol is distributed between raft and non-raft parts of the membrane almost equally [Veatch, S. L., Polozov, I. V., Gawrisch, K., & Keller, S. L. (2004). Liquid domains in vesicles investigated by NMR and fluorescence microscopy. Biophysical Journal, 86(5), 2910-2922]. A decrease in cholesterol content in membranes leads to even more robust and ordered SM-rich domains, while at cholesterol content of about 50% or higher rafts do not form at all even at room temperature (e.g., [Veatch, S. L., & Keller, S. L. (2005). Seeing spots: complex phase behavior in simple membranes. Biochimica et Biophysica Acta (BBA)-Molecular Cell Research, 1746(3), 172-185]). Appropriate range of cholesterol concentrations is substantial for rafts to have distinct physico-chemical properties (e.g., particular viscosity, lipid bilayer thickness, elastic rigidity), but cholesterol per se is not a crucial component of lipid rafts; it is a crucial component of rafts having particular set of physical properties.
6) Lines 210, 213-214: “...its positioning in the outer leaflet of the plasma membrane ...”, “...and the interaction with phosphatidylinositol-4,5-bisphosphate in the inner leaflet of the plasma membrane [63] ...”. The statement looks contradictory: SM is a lipid of the outer leaflet of plasma membrane, while PIP2 resides in the inner leaflet of plasma membranes. Their interaction seems doubtful, as SM and PIP2 do not meet. The statement should be clarified or corrected.
7) Line 348: “... has also been found that neurons prefer the liquid phase within lipid rafts”. The sentence is strange. Neuron plasma membrane is divided to liquid-ordered (rafts) and liquid-disordered phases. How can neuron (cell) prefer a particular (liquid) phase of its plasma membrane? Moreover, usually, the “liquid phase” is used as synonym for “liquid-disordered phase”. There is no liquid-disordered phase within liquid-ordered phase of rafts.
8) Line 362: “In neurons depleted of rafts ...”. Reference is required for this statement. It is not clear, how one can deplete neurons of rafts. Usually, cell plasma membranes may be depleted of cholesterol (e.g., using methyl-cyclodextrin), but this is not exactly the same as depletion of rafts. This issue should be written in a more accurate way.
9) Lines 441-442, 595-596: “...the gangliosides associate laterally with each other, sphingomyelin, cholesterol and selected proteins in lipid rafts [71].”, “... SARS-CoV-2 spike protein has a lateral N-terminal domain that binds to gangliosides residing in lipid rafts [192].” Gangliosides are not always raft components. Their distribution between raft and non-raft parts of membrane is shown to be dependent on their concentration. At physiological concentrations, gangliosides can form their own domains residing preferentially in the liquid-disordered (non-raft) part of the membrane [Bao, R., Li, L., Qiu, F., & Yang, Y. (2011). Atomic force microscopy study of ganglioside GM1 concentration effect on lateral phase separation of sphingomyelin/dioleoylphosphatidylcholine/cholesterol bilayers. The Journal of Physical Chemistry B, 115(19), 5923-5929; Galimzyanov, T. R., Lyushnyak, A. S., Aleksandrova, V. V., Shilova, L. A., Mikhalyov, I. I., Molotkovskaya, I. M., ... & Batishchev, O. V. (2017). Line activity of ganglioside GM1 regulates the raft size distribution in a cholesterol-dependent manner. Langmuir, 33(14), 3517-3524]. This issue should be clarified and discussed.
10) Lines 549-550: “A genetic deficiency in cholesterol metabolism causes Smith-Lemli-Opitz syndrome ...” Possible raft-related consequences of accumulation of metabolic precursor of cholesterol, 7-dehydrocholesterol, occurring in Smith-Lemli-Opitz syndrome are analyzed in the works [Staneva, G., Chachaty, C., Wolf, C., & Quinn, P. J. (2010). Comparison of the liquid-ordered bilayer phases containing cholesterol or 7-dehydrocholesterol in modeling Smith-Lemli-Opitz syndrome [S]. Journal of lipid research, 51(7), 1810-1822; Staneva, G., Osipenko, D. S., Galimzyanov, T. R., Pavlov, K. V., & Akimov, S. A. (2016). Metabolic precursor of cholesterol causes formation of chained aggregates of liquid-ordered domains. Langmuir, 32(6), 1591-1600].
11) Lines 556-558: “When incorporated into lipid rafts, they increase the content of unsaturated fatty acids and alter the physical properties of lipid rafts by affecting their fluidity and stability ...” Lipids with unsaturated, and, especially, polyunsaturated chains are strongly excluded from rafts; they do not incorporate into rafts. Polyunsaturated lipid can affect raft properties, but not via incorporation into them: they alter the properties of the surrounding membrane that results in alteration of the structure of raft-surrounding membrane boundary, that, in turn, affects rafts. In particular, the presence of polyunsaturated lipids in the surrounding membrane leads to an increase of the energy of the interphase boundary (between liquid-ordered (raft) and liquid-disordered phases). This increase in the boundary energy affects e.g. the size distribution of rafts.
Author Response
Dear Reviewer 2,
Thank you for taking the time to review our manuscript. Your insightful comments and feedback are greatly appreciated, and we are grateful for the opportunity to address them.
We believe that your comments provide valuable insights that will help strengthen the quality and clarity of our work.
We have carefully reviewed each of your comments and suggestions, and we are pleased to provide our responses and proposed revisions below. Your feedback is instrumental in refining our research and ensuring its significance in the field.
Answers are attached.
Sincerely,
Authors
